# Anteroposterior Ethmoidectomy in the Endoscopic Reduction of Medial Orbital Wall Fractures: Does It Really Reduce Stability?

Antonio Romano [1], Stefania Troise [1], Francesco Maffia [1], Umberto Committeri [1,*], Lorenzo Sani [1], Marco Sarcinella [1], Antonio Arena [1], Giorgio Iaconetta [2], Luigi Califano [1] and Giovanni Dell'Aversana Orabona [1]

[1] Maxillofacial Surgery Unit, Department of Neurosciences, Reproductive and Odontostomatological Sciences, "Federico II" University of Naples, Via Pansini 5, 80100 Naples, Italy

[2] Neurosurgery Unit, Department of Medicine, Surgery and Odontoiatrics, University of Salerno, Via Giovanni Paolo II 132, 84084 Salerno, Italy

* Correspondence: umbertocommitteri@gmail.com

**Abstract:** The surgical treatment of isolated medial orbital wall fractures is still a much-debated topic in the literature due to the choice of many surgical accesses. The main options are represented by transcutaneous versus endonasal endoscopic approaches. Our study aims to clarify the role of ethmoidectomy in the pure endoscopic endonasal reduction of medial orbital wall fractures, evaluating the immediate postoperative outcome and its long-term stability. A total of 31 patients affected by isolated medial orbital wall fracture, treated only by endoscopic approach, were included in the study and divided in two groups: (A) 14 patients treated by endoscopic reduction and anterior ethmoidectomy; (B) 17 patients treated by endoscopic reduction and anteroposterior ethmoidectomy. Perioperative and 6-month postoperative follow-up CT scans were performed. With the use of 3D medical software, we evaluated the comparison between the treated orbit and the mirrored contralateral orbit in the two groups, in order to observe the reduction of the fracture. Furthermore, to check the stability of reduction and to evaluate any medial orbital wall changes, we provided a comparison between the 3D CT scan orbital images of immediate postoperative CT and 6-month follow-up. Data obtained showed that the intraoperative surgical reduction was successful in all 31 cases, but it was better in Group B. Stability of the reduction at 6 months was observed in both groups without significant discrepancies. In our opinion, the endonasal endoscopic approach with ethmoidectomy represents a valid and useful technique by which to treat medial orbital wall fractures. The anatomical detail of the buttressing structures of the medial orbital wall, as the second portion of the middle turbinate, grants long-term stability of the surgical outcome.

**Keywords:** medial orbital wall fracture; traumatology; endoscopic approach; ethmoidectomy; endoscopic fracture reduction; CAD/CAM maxillofacial surgery

## 1. Introduction

Isolated medial wall fracture is the most common type of orbital blowout fracture due to the thickness of the lamina papyracea, which is a very thin bony plate (0.2–0.4 mm thick) [1]. Many different techniques have been proposed in order to approach and manage medial wall fractures, such as a Lynch incision that has today been abandoned because it leads to poor aesthetic results; the subciliary incision, which provides a limited view, with the additional cost of a skin incision; and the transconjunctival approaches, which include the transcaruncular, precaruncular, and retrocaruncular [2–4]. Some authors have applied a combined approach, adopting endoscopic assistance during the placement of reconstruction material for better positioning accuracy [5]. In terms of the choice of surgical approach, several authors have also considered the anatomy of the structures involved in the fracture. If the fracture does not involve the inferomedial corner of the orbit (inferomedial orbital strut—IOS), which represents

a stable structure of the orbit, the endoscopic approach can also be used unaccompanied [6]. The endonasal endoscopic approach, which was first introduced by Yamaguchi et al. in 1991, is today considered a fundamental instrument with which to treat medial orbital wall fractures; however, how to perform it still represents a debated topic in the literature [7]. Several authors have performed the removal of the uncinate process and the ethmoid bulla to increase visibility during the procedure [8,9]; others have affirmed that the ethmoid bulla, the middle turbinate lamina basalis, and/or the uncinate process may act as a buttress for the medial orbital wall [4]. Therefore, the loss of these structures may alter the biomechanics of the medial wall and reduce stability [4,10]. The aim of our study is to clarify the role of anteroposterior ethmoidectomy in the endoscopic endonasal reduction of pure medial orbital wall fractures by evaluating the immediate postoperative outcome and its long-term stability.

## 2. Materials and Methods

### 2.1. Patient Selection

The study was prospectively performed on patients who underwent endoscopic reduction of medial orbital wall fractures in the Maxillofacial Surgery Unit of the University Federico II of Naples from January 2019 to February 2022. The study was conducted following the principles of the Declaration of Helsinki. Written informed consent was obtained from each patient after submission along with the approval of the Ethics Committee of the University, with protocol number 370/2019. Patients with the following inclusion criteria were enrolled:

- Patients age > 18 years old;
- Patients with a medial orbital wall fracture, shown by coronal and axial CT scans, not involving the inferomedial angle of the orbit;
- Patients with combined orbital walls fractures, without the involvement of zygomatic complex;
- Patients with fractured bony fragments displaced;
- Patients who did not require urgent surgery;
- Patients treated only by endoscopic approach;
- Patients with functional defects (diplopia, eye motility limitation, enophthalmos).

A total of 48 patients with a diagnosis of orbital medial wall fracture were considered: 13 of them were excluded because of the inferomedial angle involvement; 3 were not selected due to the absence of functional defects and no surgical indications. One was excluded due to a previous skin laceration. A total of 31 patients were included in the study according to the above inclusion criteria. The 31 patients were divided in two groups: (A) 14 patients treated between January 2016 and November 2018, when only anterior ethmoidectomy was performed; (B) 17 patients treated between December 2018 and December 2021, after the introduction of anteroposterior ethmoidectomy in the surgical procedures. For all of the 31 patients, the same 3D digital workflow and analysis was performed. The primary purpose was to prove that the anteroposterior ethmoidectomy surgical procedure improved the view of the operative field, obtaining a better reduction of the fracture, particularly in the posterior portion. The second purpose was to demonstrate that this procedure does not reduce the stability of the medial orbital wall over time. Furthermore, the most recent retrospective studies on this topic in the literature from 2010 were analyzed to obtain a brief review.

### 2.2. Presurgical Evaluation

The collected data included age, sex, and cause of trauma. Clinical evaluation was performed by an ophthalmologist and involved diplopia, enophthalmos, and eye motility limitation. All patients underwent a CT scan within 12 h after the trauma, and then the images were analyzed to identify the medial orbital wall fracture side.

### 2.3. Surgical Procedure

All 31 patients underwent surgery within 4 days after the trauma. All the procedures were performed under general anesthesia by the same expert surgeon using a 4-mm, 0–30°

endoscope. Cottonoids soaked with diluted epinephrine (1:100,000) were filled in the middle meatus. A solution of 1:100,000 epinephrine and 2% lidocaine was injected into the anterior border of the middle turbinate, around the uncinate process, and into the adjacent septum and lateral wall of the middle turbinate. The middle turbinate was gently medialized to improve the operating field. The surgical procedure was performed as follows:

- Group A: A horizontal incision was performed over the uncinate process, allowing better exposition, and it was then excised to access the ethmoid bone. An anterior ethmoidectomy was performed by removing the ethmoidal bulla to individuate the fracture site and, eventually, the herniated orbital tissue. The fracture was reduced by blunt dissection using a Cottle dissector.
- Group B: The procedure included the same as Group A, with additional posterior ethmoidectomy. The second portion of the middle turbinate, the lamina basalis, was crossed in its superomedial segment to expose the posterior portion of the lamina papyracea. After identifying the site of the medial orbital wall fracture, the herniated orbital contents were pushed into the orbit by a Cottle dissector. The medial wall was gently repositioned with a peristome following the anatomy. The surgical procedure is shown in Figure 1.

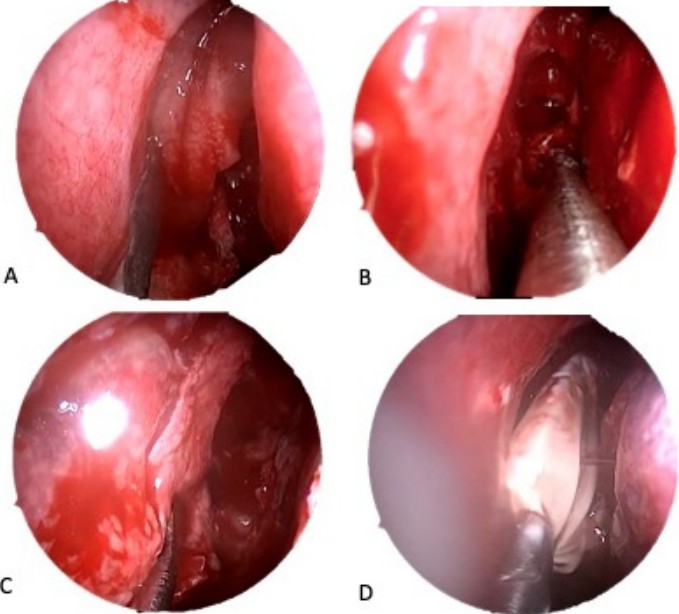

**Figure 1.** Surgical Procedure: (**A**) uncinectomy; (**B**) bullectomy; (**C**) fracture reduction; (**D**) splint positioning.

In both groups, to maintain the reduction, the fracture was contained with a small gloved nylon-anchored Merocel that remained in place for a period of 4–7 days. A long Merocel was positioned in the treated nasal fossa and then removed after 3 days.

### 2.4. Post-Surgical Immediate Evaluation

After discharge, patients started a 3-week protocol with nasal rinse to maintain proper cleaning of the nasal fossae. Weekly endoscopic controls were performed to check the correct healing of the treated fossa. All 31 patients were studied in the postoperative phase with a CT scan the day after the surgery. The digital workflow was based on: (1) acquisition of CT DICOM files into the software Invesalius (Version 3.1.1), to obtain the 3D image of the patient's scan; (2) acquisition of the 3D image in the software MeshMixer (Version 3.5.474), an interactive segmentation tool that used a 3D texture–painting interface for bone separation. The segmentation function "Plane cut" allowed for rapid separation of the single bone fragments of the orbits for the segmentation of the entire orbit. Manual segmentation was performed

for the affected orbit after the reduction, and for the contralateral orbit. With the function "Mirroring", a specular image of the contralateral healthy orbit was obtained. (3) Comparison of the two 3D orbital images with the software Geomagic Design X (Version 2019.0.2): the functions "Alignment scans" and "Accuracy Analyzer" allowed the creation of a distance color map to compare the specular healthy orbit image and the post treatment orbit image, underlining possible discrepancies. The discrepancy between overlapping model points was visualized through a color map fusion image for all 31 cases treated. The upper/lower limit for color coding the discrepancies was fixed as +1.5 mm and −1.5 mm, so deviations appeared in different colors. We used steps of 0.15 mm, so each color encoded a distance interval of 0.15 mm. The discrepancy between the two orbits images for all 31 patients was computed at nine medial orbital wall points: Point A, anterosuperior angle; Point B, anteromedian angle; Point C, anteroinferior angle; Point D, middle-superior angle; Point E, middle median angle; Point F, middle inferior angle; Point G, posterosuperior angle; Point H, posteromedial angle; Point I, posteroinferior angle.

The workflow process is shown in Figures 2–5.

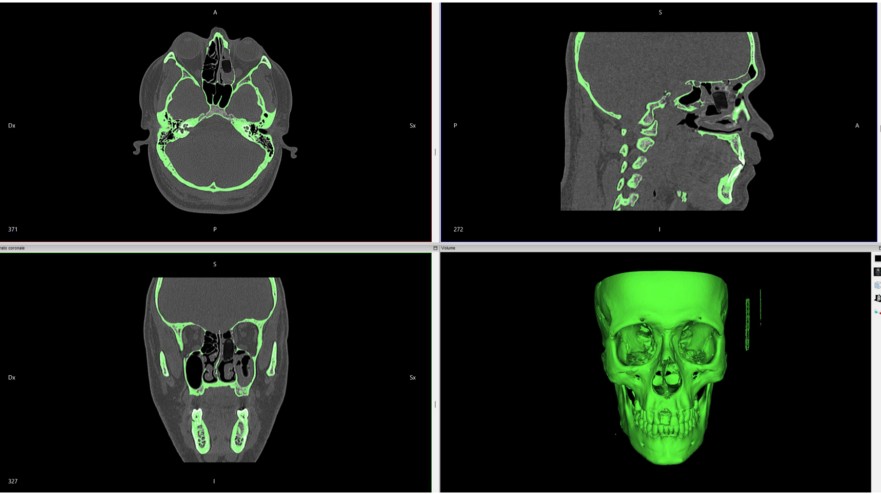

**Figure 2.** Invesalius software digital workflow. The imported file DICOM is transformed into a 3D image.

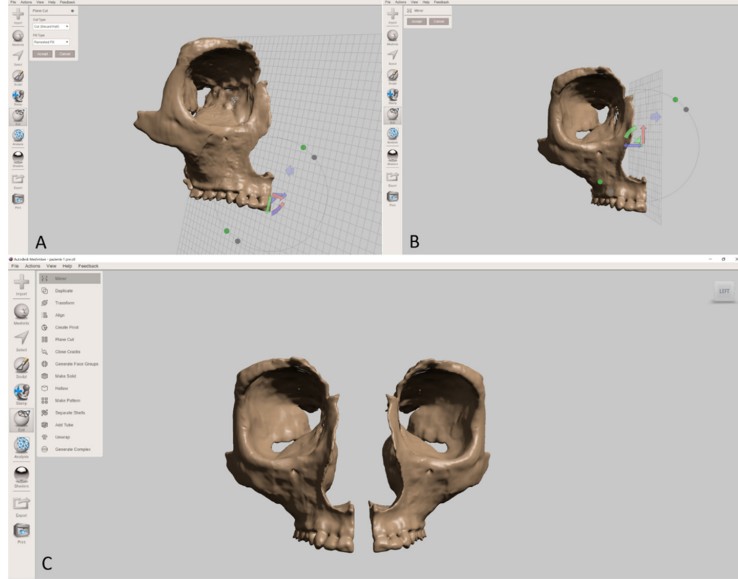

**Figure 3.** Meshmixer software digital workflow. (**A**) Manual segmentation of the affected orbit; (**B**) mirroring process; (**C**) final mirrored healthy orbit.

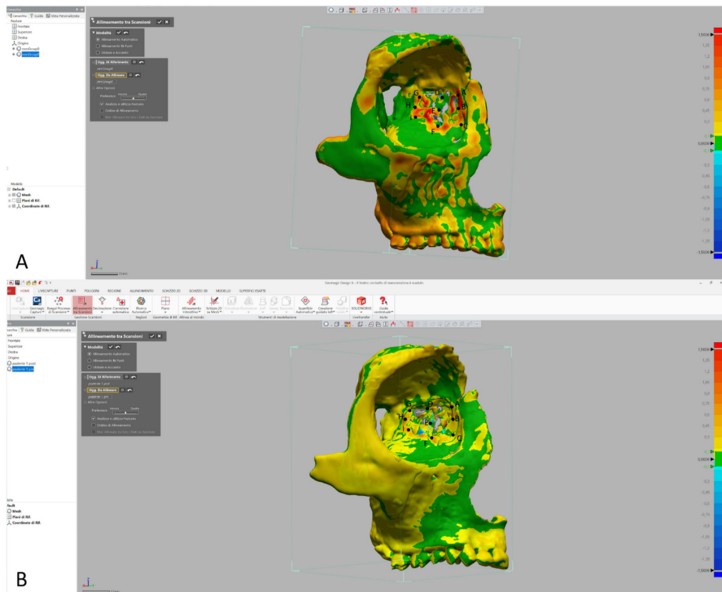

**Figure 4.** Color map distance between specular healthy orbit and affected orbit post treatment in the immediate postoperative time in Geomagic Design X software with the nine anatomical points considered. (**A**) Group A patient. (**B**) Group B patient.

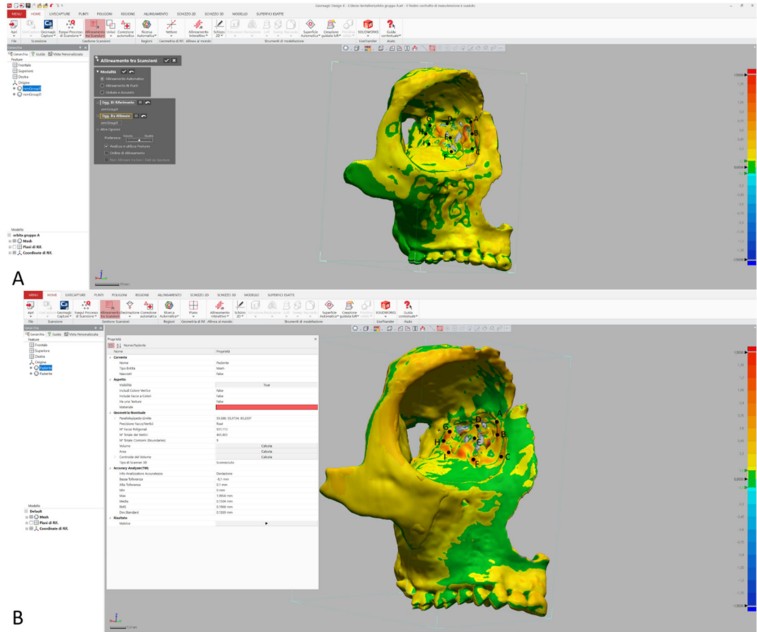

**Figure 5.** Color map distance between immediate post treatment orbit and 6-month follow-up orbit in Geomagic Design X software with the nine anatomical points considered. (**A**) Group A patient. (**B**) Group B patient.

*2.5. Follow-Up Evaluation*

Clinical examination in terms of diplopia, enophthalmos, and eye motility limitation was evaluated in outpatient visits by an ophthalmologist. All 31 patients underwent a CT scan at the 6-month follow-up. The 3D digital workflow was based on: (1) acquisition of CT DICOM files into the software Invesalius (Version 3.1.1) to obtain the 3D image of the patient's scan; (2) acquisition of the 3D image in the software MeshMixer (Version 3.5.474) to allow the segmentation of the entire orbit. (3) Comparison with the software Geomagic Design X (Version 2019.0.2) of the 3D immediate post-operative orbit image, and 3D 6-month postoperative orbit image: a distance color map was obtained to compare the

two orbits images underlining possible discrepancies. The setting was the same and the nine anatomical points were the same.

## 3. Results

### 3.1. Clinical Results

The main features of the 31 patients are shown in Table 1.

**Table 1.** Main features of 31 patients enrolled in the study. Legend: D: diplopia; E: enophthalmos; EML: eye motility limitation; A: Group A patient; B: Group B patient.

| Case | Age | Sex | Cause of Trauma | Side of Fracture | Clinical Presurgical Examination | Clinical Postsurgical 6 Months Examination |
|------|-----|-----|-----------------|------------------|-----------------------------------|--------------------------------------------|
| 1 A | 45 | M | Car accident | Right | D | - |
| 2 A | 27 | M | Moto accident | Left | E + EML | - |
| 3 A | 56 | F | Falling | Left | D | - |
| 4 A | 35 | M | Moto accident | Left | D + E | - |
| 5 A | 21 | M | Sport accident | Left | E + EML | - |
| 6 A | 35 | M | Car accident | Right | D + E | D |
| 7 A | 47 | F | Falling | Left | E | - |
| 8 A | 55 | M | Fighting | Left | D + EML | D |
| 9 A | 68 | M | Falling | Right | D + E | - |
| 10 A | 19 | M | Sport accident | Right | E | - |
| 11 A | 25 | F | Moto accident | Left | D | - |
| 12 A | 76 | F | Falling | Right | D + E + EML | - |
| 13 A | 61 | M | Car accident | Right | D + E | - |
| 14 A | 40 | F | Car accident | Left | D | - |
| 15 B | 63 | F | Falling | Left | D | - |
| 16 B | 49 | M | Falling | Left | D | - |
| 17 B | 58 | M | Car accident | Right | D | D |
| 18 B | 52 | M | Fighting | Right | E | - |
| 19 B | 80 | F | Car accident | Left | D + E | - |
| 20 B | 48 | F | Car accident | Right | D + E | - |
| 21 B | 40 | F | Fighting | Right | E | - |
| 22 B | 36 | M | Car accident | Right | D + E | - |
| 23 B | 18 | M | Sport accident | Left | EML | - |
| 24 B | 45 | M | Car accident | Right | E | - |
| 25 B | 37 | M | Moto accident | Left | D + E | D |
| 26 B | 72 | F | Falling | Left | E + EML | - |
| 27 B | 66 | F | Falling | Right | D + E + EML | - |
| 28 B | 48 | F | Car accident | Left | D + E | - |
| 29 B | 31 | M | Sport accident | Left | D + E | - |
| 30 B | 41 | M | Fighting | Left | E + EML | - |
| 31 B | 68 | F | Car accident | Right | D + E | - |

Group A: The mean age of the patients was 43.5 (interquartile: 19–80), and the M:F sex ratio was 1.8:1. The most affected side was the left side (8/14, 57.1%). The most frequent causes of trauma were car accidents and falling (both 4/14, 28.5%), followed by motorcycle accidents (3/14, 21.4%) and sport accidents (2/14, 14.2%). Only one case of fracture due to fighting was recorded. Diplopia was the most referred symptom among cases (10/14, 71.4%), followed by the clinical evidence of enophtalmos (9/14, 64.25%). Eye movement limitation was observed in four cases (28.5%). A combination of diplopia and enophtalmos was recorded in five cases (35.7%). Only one patient experienced all three findings. In the postoperative stage, only two cases of diplopia were referred at the one month control, while no enophthalmos and eye motility limitations were recorded during the postoperative follow-up period. Regarding the surgery, the average operative time was 45 min (range: 40–65).

Group B: The mean age of the patients was 50 (interquartile: 18–80), and the M:F sex ratio was 1.125:1. The most affected side was the left side (9/17, 52.9%). The most frequent cause of trauma was car accident (7/17, 41.1%), followed by falling (4/17, 23.5%) and fighting (3/17, 17.6%). Only two cases by sport accident and one by motorcycle accident were recorded. Enophtalmos was the most observed clinical finding (13/17, 76.4%), followed by diplopia (11/17, 64.7%) and eye movement limitation (4/17, 23.5%). In eight cases, diplopia was associated with enophtalmos (8/17, 47%). Only one patient experienced all three findings. In the postoperative phase, only two cases of diplopia were referred at the one month control, while no enophthalmos and eye motility limitations were recorded during the postoperative follow-up period. Regarding the surgery, the average operative time was 75 min (range: 50–110).

### 3.2. Immediate Comparison Results

Group A: For these fourteen patients, discrepancy between the nine anatomical points considered was more than 1.5 mm, with an overall average of 1.95 mm. All averages were over 1.5 mm. The highest discrepancy was observed at Point I with 2.96 mm; the lowest was at Point F with 1.02 mm. The highest average was 2.27 mm, recorded at Point G, while the lowest was 1.73 mm, recorded at Point F. The posterior border points averages were all over 2 mm of discrepancy: G (2.27 mm), H (2.01 mm), and I (2.21 mm).

Group B: For these seventeen patients, discrepancy between the nine anatomical points considered was less than 1.5 mm, with an overall average of 1.11 mm. All averages were under 1.5 mm. The highest discrepancy was observed at Point H, with 2.17 mm; the lowest was at Point C, with 0.36 mm. The highest average was 1.21 mm, recorded at Point H, while the lowest was 0.92 mm, recorded at point F. No particular trends were observed between anterior, middle, and posterior borders.

The complete values of discrepancy for all patients of Group A and Group B are shown in Figures 6 and 7, respectively.

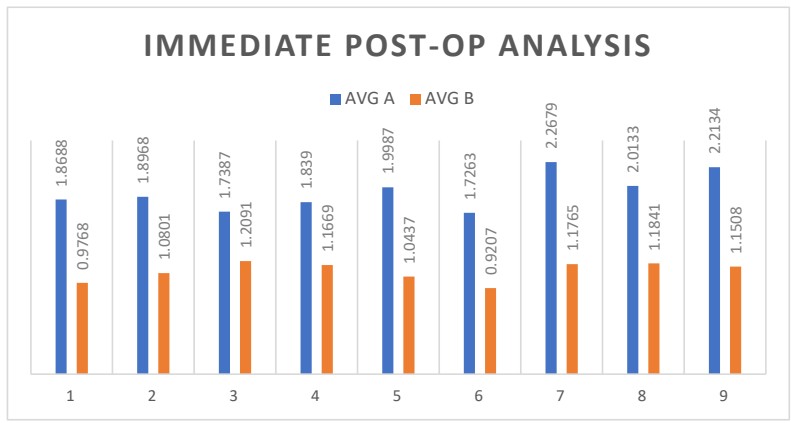

**Figure 6.** Discrepancy values of nine medial orbital wall points between specular healthy orbit image and affected orbit immediately post treatment for Group A compared with Group B.

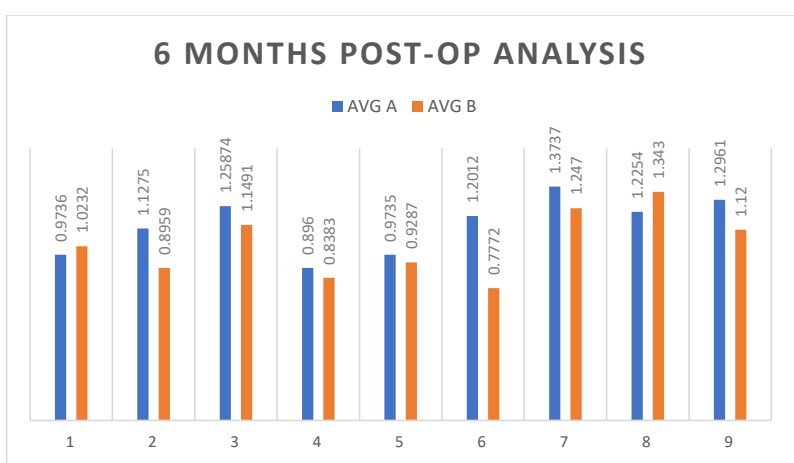

**Figure 7.** Discrepancy values of nine medial orbital wall points between immediately post treatment affected orbit image and affected orbit at 6 months post treatment for Group A compared with Group B.

*3.3. Follow-Up Comparison Results*

Group A: For these fourteen patients, the discrepancy between the nine anatomical points considered was less than 1.5 mm, with an overall average of 1.14 mm. All averages were under 1.5 mm. The highest discrepancy was observed at Point H, with 2.03 mm; the lowest was at Point A, with 0.47 mm. The highest average was 1.38 mm, recorded at Point G, while the lowest was 0.90 mm, recorded at Point D.

Group B: For these seventeen patients, the discrepancy between the nine anatomical points considered was less than 1.5 mm, with an overall average of 1.02 mm. All averages were under 1.5 mm. The highest discrepancy was observed at Point H, with 1.78 mm; the lowest was at Point F, with 0.26 mm. The highest average was 1.34 mm, recorded at Point H, while the lowest was 0.78 mm, recorded at Point F.

## 4. Discussion

Orbital wall fractures are very common because of the weakness of the bony structures composing the orbit [3]. In particular, the medial orbital wall is the weakest wall: the lamina papyracea is described as having the thinnest structure (0.2–0.4 mm) and, for this reason, it represents the most fractured orbital wall [1]. The biomechanics of this kind of fracture consists of two theories: the buckling theory, where the force is absorbed and transmitted from the orbital rim to the surrounding anatomical structures, until a point of weakness is encountered, such as the medial orbital wall; and the hydraulic theory, which describes how the force is transferred from the globe to the orbit until the medial orbital wall becomes fractured [11]. In our population, 50% of cases were victims of high kinetic energy events, such as a traffic accident, in line with the trend described in the literature. Causes of trauma such as fighting and sportive accidents were reported in the younger population of the study, while traumas originated from accidental falls involved older patients. We observed that the most significant clinical signs and symptoms, such as diplopia and enophthalmos, derived from traffic accidents, supporting the force transmission theories.

The surgical treatment of isolated medial orbital wall fractures is still a much debated topic in the literature because of different surgical access availability [12]. The most used approaches are the cutaneous approaches, such as the subciliary, the transcaruncular, the "Lynch" approach, with or without endoscopic assistance, and the pure endoscopic approach with transethmoidal endonasal access [3,6]. The gradual development and diffusion of endoscopic surgery has reduced the invasiveness of classic surgical procedures and improved aesthetic outcomes, especially in maxillofacial traumatology [5,7]. Endoscopic assistance in cutaneous approaches improves the operating field view and allows for checking the correct positioning of the selected repairing material. [13]. The advantages of the pure endoscopic approach are represented by the absence of external skin scars, a close and direct visualization of the fracture site, and

the possibility of not implanting alloplastic material [14,15]. The most significative limit to this procedure is represented by the long learning curve. In our casuistry, the anteroposterior ethmoidectomy was performed after 3 years of experience. For this reason, it was possible to consider two different populations, with and without direct posterior visualization of the medial wall. To reach the fracture, particularly if it is sited behind the lamina basalis of the middle turbinate, it is necessary to pass through some important structures, as normally performed during FESS/ESS surgery. The lamina papyracea is normally encountered on the way to the sphenoid sinus. For this reason, in this approach, the medial orbital wall can be considered a key point, being normally identified during sinus surgery.

The anatomical buttressing of the lamina papyracea is unique: despite its thinness, many ethmoidal cell septa perpendicularly support the wall [4]. Another important supporting structure is represented by the middle turbinate (MT), a crucial landmark in the identification of the skull base, ethmoid cells, and lamina papyracea in functional endoscopic sinus surgery (FESS) [7,16]. The MT is composed of an intranasal part, called the "lamina recurved", and an intraethmoidal lamina, the lamina basalis. This second structure describes the entry zone for the posterior ethmoid: to pass gently through this structure plays a key role in maintaining structure long-term stability [9].

The most recent literature was examined to evaluate the different surgical techniques adopted. The search was conducted on Pubmed, with a retrospective limit of 10 years. Only papers regarding isolated medial orbital wall fractures were included in the evaluation. Several authors have described the importance of endoscopic surgery, either alone or combined with an external approach, as shown in Table 2.

For each paper, we analyzed the endoscopic surgical steps and the implanted material. The analysis of the literature showed that the endoscopic endonasal approach has become the gold standard for this kind of fracture. All authors reported the apposition of alloplastic material, such as fine sheets of Medpor (Stryker, Kalamazoo, MI, USA), Synpor (Depuy Synthes; Johnson and Johnson, New Brunswick, NJ, USA), dura mater substitutes, or a combination of a titanium mesh with a layer of porous Medpor [4,12,14,17]. Four papers described anteroposterior ethmoidectomy, performed for most posterior fractures as a necessary step to dominate the fracture field [18,19]. The authors of all the selected papers adopted the apposition of alloplastic sheets on the fractured lamina [10,14,20–22]. In one of these four papers, it reported the use of intraoperative navigational assistance to obtain a more precise reduction by uploading a mirroring of the healthy orbit. Navigation was also used to compare orbital volumes before and after treatment, to assess the stability of the outcome [8].

The surgical procedure we performed was in agreement with the data obtained from the literature regarding surgical approaches: in our procedure, not considering if the fracture was anterior or posterior, the medial orbital wall was accessed by removing the uncinate process, the ethmoid bulla, and passing through the lamina basalis of the middle turbinate, facing the sphenoid sinus as the posterior limit. This corridor allowed us to increase the visibility of the surgical field and to completely dominate the fracture. This passage must be carefully performed, to respect the locoregional buttressing. When the MT buttress is preserved, the postsurgical anatomy remains intact in the long term, resulting in an excellent outcome. In our cases, no implantation of absorbable material was performed. Despite the fact that the width of the fracture could represent an indication, we preferred a more conservative approach; in the larger fracture, we did not remove bony fragments. We just gently laterally pushed them back. Once the fractured wall was reduced, a small gloved nylon-anchored Merocel was used as a temporary endonasal splint to maintain the position of the fragments. Keeping the small Merocel gloved preserves it from fluid imbibition that could decrease its rigidity. This technique has been successful in terms of both clinical results in the immediate postoperative period and after 6 months. Compared to the literature, the adoption of the endoscopic approach cancels the incidence of complications, such as scarring, lagophtalmos, and epiphora, risks associated with the transcutaneous approach. The decision to not insert alloplastic material reduces the

incidence of complications such as implant infection, residual diplopia, and material migration [13]. In our population, cases of residual diplopia (four patients; two per group) were evaluated by immediate postoperative clinical imaging in order to assess eventual reduction of misplacement. Incidence was attributed to postoperative edema and did not undergo further treatments other than corticosteroid therapy. On the basis of our experience, clinical outcome improved compared to patients treated with the previous transcutaneous approach. We recorded the resolution of preoperative symptoms in almost all cases and the stability of the anatomical reduction of the fracture after 6 months.

**Table 2.** Most recent literature regarding the surgical procedure of medial orbital wall fracture reduction. Approach: EE—endoscopic endonasal; EAs—endoscopic assistance; RCa—retrocaruncular; TCa—transcaruncular; PCa—precarunculare; SP—subpalpebral; TCv—trancongiuntival. Surgical Technique: UC—uncinectomy, MSA—maxillary sinus antrostomy, BC—bullectomy, AE—anterior ethmoidectomy, APE—anteroposterior ethmoidectomy, pMTT—middle turbinate turbinectomy, MTT—middle turbinate turbinectomy, MTL—middle turbinate luxation, ORIF—open reduction, internal fixation.

| Author | Year | Approach | Surgical Technique | Implants |
|---|---|---|---|---|
| Park et al. [10] | 2009 | EE | UC, BC, AE | Silastic sheet |
| KyoungHoon Kim et al. [14] | 2010 | EE | UC, BC, AE | Silastic sheet |
| Gerbino et al. [17] | 2015 | RCa, EAs | ORIF | Titanium mesh |
| Copelli et al. [8] | 2015 | EE | UC, BC, AE/APE | Silastic sheet |
| Pagnoni et al. [22] | 2015 | EE/SP | ORIF | Medpor sheet |
| Colletti et al. [3] | 2016 | EE | APE | Medpor sheet |
| Kun Hwang et al. [4] | 2016 | EE | UC, BC, APE | Medpore sheet |
| Procacci et al. [18] | 2016 | EE | UC, MSA, BC, APE | Dura mater substitutes, Medpore |
| Seong Hwan Bae et al. [10] | 2019 | EE | pMTT, AE | Titanium mesh with Medpor |
| Bonsembiante et al. [20] | 2019 | EE | UC, AE | Medpor with fibrin glue, Endonasal splint |
| Dong et al. [21] | 2019 | TCa | ORIF | Unsitered Hydroxyapatite Particle/Poly L-lactide sheet |
| Taewoon Kim et al. [16] | 2020 | EE vs TCa | UC, MSA, AE, MTL | Synpor sheet |

The goal of personalized patient care has led to the growing use of new technologies and dedicated software [23,24]. Our 3D digital analysis showed that the discrepancy between the mirrored image of the untreated and the treated orbit was generally always greater in Group A, with higher values recorded in the most posterior points of reference. This event probably indicated that the surgical procedure of the anteroposterior ethmoidectomy allows for a better view of the posterior portion of the medial orbital wall, thus confirming a more precise fracture reduction. The second comparison, immediate postoperative versus 6 months, showed how, despite the anteroposterior ethmoidectomy, stability results are comparable between the two groups, without particular trends between the three zones. The most valuable benefit of the APE resides in the complete exposition of the lamina papyracea that grants a wider domination of the fracture, even in the more anterior cases. As suggested by the 3D analysis, reduction was more stable in the long term with a minimal displacement.

Considering the length of hospitalization, we did not observe a clear difference between the two groups: a mildly longer hospitalization, ranging from +1 and +2 days, was observed

in Group B, due to the removal of the splint on the fourth/fifth day. For these reasons, the institutional cost of both procedure is comparable.

This study aimed to investigate the best surgical procedure by which to treat medial orbital wall fractures by an endoscopic only approach, with a focus on the peculiarity of the surgical technique and the encountered anatomy. The evaluation performed included only objective analysis, such as clinical presentation and 3D analysis. Further studies on a larger scale are necessary and should also include a procedure comparison that assesses patient-related status and satisfaction. The limitations of our study can be summarized by the following: sample size of the population; single-center investigation; difficult reproducibility of the study due to the adoption of software requiring long learning curve.

Our technique demonstrates how the endoscopic approach represents a safe and standardized procedure for the reduction of orbital medial wall fractures. Respecting the regional anatomy, in particular during the passage through the lamina basalis, means also respecting the normal buttressing of the naso-orbital region, avoiding the weakening of the involved bone structures. The 3D digital analysis after the surgical treatment shows that, although ethmoidectomy is performed, the reduction of the fracture remains stable over time.

## 5. Conclusions

The endonasal endoscopic approach represents a valid and useful technique by which to treat medial orbital wall fractures. Although anteroposterior ethmoidectomy is performed to improve visibility and dominate the fracture, anatomical respect of the buttressing structures of the medial orbital wall, such as the second portion of the middle turbinate, grants long-term stability of the surgical outcome.

In conclusion, we can affirm that, during the endoscopic reduction of medial orbital wall fractures, it is useful to perform anteroposterior ethmoidectomy to improve the visibility and repositioning of fractured fragments without the risk of decreasing the stability of the reduction; being careful not to remove important structures such as the entire second portion of the middle turbinate.

**Author Contributions:** Conceptualization, A.R. and L.C.; methodology, S.T.; software, U.C.; validation, A.R., G.D.O. and L.C.; formal analysis, A.A.; investigation, F.M. and M.S.; resources, F.M.; data curation, M.S.; writing—original draft preparation, F.M.; writing—review and editing, S.T.; visualization, L.S.; supervision, A.R. and G.I.; project administration, L.C. All authors have read and agreed to the published version of the manuscript.

**Funding:** This research received no external funding.

**Institutional Review Board Statement:** The study was conducted in accordance with the Declaration of Helsinki, and approved by the Institutional Review Board (or Ethics Committee) of Federico II University of Naples (protocol code 370/2019 and date of approval 21 February 2020).

**Informed Consent Statement:** Informed consent was obtained from all subjects involved in the study. Written informed consent has been obtained from the patient(s) to publish this paper.

**Conflicts of Interest:** The authors declare no conflict of interest.

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
