# Peer review of "Anteroposterior Ethmoidectomy in the Endoscopic Reduction of Medial Orbital Wall Fractures: Does It Really Reduce Stability?"

_applsci, doi:10.3390/app13010098_

Round 1
Reviewer 1 Report
I would like to thank the authors for all their work, it addresses several knowledge gaps in medial orbital wall repair, to further address those questions:
Can the authors comment on their clinical outcomes (both with anterior and anterior posterior approaches) compared to the cited literature utilizing various implants specifically did it appear to be statistically equal or improved?
Can the authors further discuss the value of the anterior posterior approach in all cases if it appears that while it provides a statistically significant reduction the two approaches provide a clinically equivalent result, especially where the anterior approach on average took an additional 30 minutes.
Is there a specific location of the fracture that would dictate a certain approach?
Reviewer 2 Report
They should include the limitations of the study, in my opinion.
Author Response
2) Can the authors further discuss the value of the anterior posterior approach in all cases if it appears that while it provides a statistically significant reduction the two approaches provide a clinically equivalent result, especially where the anterior approach on average took an additional 30 minutes.
Is there a specific location of the fracture that would dictate a certain approach?
2) We think that a greater sample would emphasize the clinical differences among the two techniques. Considering the outcomes obtained, our approach is to continue to perform APE as it offers better long term results and is affected by a minimal displacement. This difference could not be appreciable considering AVGs, but in singular values.
We tried to understand if the fracture location could be a checkpoint for one of the two procedures. The answer is No, because the posterior edge (highest point of PO vs PO-6m misplacement) of the fracture can be dominated only by APE.
Reviewer 3 Report
Thank you very much for the chance to review the present manuscript. The authors performed a well-designed comparison of anterior versus anterioposterior ethmoidectomy in the endoscopic reduction of orbital wall fractures. To compare the outcomes, the authors assessed radiological parameters as well as complication rates up to 6 months follow-up.
I have the following minor comments:
1)the operation time was longer in group B. Please elaborate on how this could affect institutional cost over time as well as patient outcomes based on your experience.
2)please include a limitation section discussing the limitations of your study. This could, for example, include that you did not assess patient-related outcome measures, which could be potentially helpful for the comparison of interventions. Further, it would be helpful to assess other potentially clinically relevant parameters in future, such as length of hospital stay, which might differ due to the more invasive in the anteroposterior intervention.
3)It is hard to compare the discrepancy results in the way you presented them in the manuscript. It would be helpful to include a box plot or bar graph showing Group A versus Group B in one graph according to the variables “AVG A” and “AVG B”. In this way, it is easier for readers to compare the discrepancy results between the two interventions.
Author Response
Reviewer 3
I have the following minor comments:
1)the operation time was longer in group B. Please elaborate on how this could affect institutional cost over time as well as patient outcomes based on your experience.
1) Line 397
2a) please include a limitation section discussing the limitations of your study. This could, for example, include that you did not assess patient-related outcome measures, which could be potentially helpful for the comparison of interventions.
2b) Further, it would be helpful to assess other potentially clinically relevant parameters in future, such as length of hospital stay, which might differ due to the more invasive in the anteroposterior intervention.
2a) Lines 399-406, Lines 395-398
2b) Line 395, The invasiveness is only technical and not affects critically the clinical state, the hospitalization parameters are in line with the other FESS interventions which have similar hospitalizations regardless of the open sinuses, except for the frontal one.
3) It is hard to compare the discrepancy results in the way you presented them in the manuscript. It would be helpful to include a box plot or bar graph showing Group A versus Group B in one graph according to the variables “AVG A” and “AVG B”. In this way, it is easier for readers to compare the discrepancy results between the two interventions.
3) New bar graphs based on AVGs has been added to the text.
